# Velvet Family Members Regulate Pigment Synthesis of the Fruiting Bodies of *Auricularia cornea*

**DOI:** 10.3390/jof9040412

**Published:** 2023-03-27

**Authors:** Xiaoxu Ma, Lixin Lu, Youmin Zhang, Ming Fang, Kaisheng Shao, Xu Sun, Fangjie Yao, Peng Wang

**Affiliations:** 1Engineering Research Centre of Chinese Ministry of Education for Edible and Medicinal Fungi, Jilin Agricultural University, Changchun 130118, China; 2Lab of Genetic Breeding of Edible Mushroom, Horticultural, College of Horticulture, Jilin Agricultural University, Changchun 130118, China; 3Economic Plants Research Insitute, Jilin Academy of Agricultural Sciences, Changchun 130118, China

**Keywords:** *Auricularia cornea*, pigment, genetic population, candidate genes, Velvet family

## Abstract

Color is a crucial feature to consider when breeding and improving strains of *Auricularia cornea*. To uncover the mechanism of white strain formation in *A. cornea*, this study selected parental strains that were homozygous for the color trait and analyzed the genetic laws of *A. cornea* color through genetic population construction, such as test-cross, back-cross, and self-cross populations, and the statistical analysis of color trait segregation. Moreover, the study developed SSR molecular markers to construct a genetic linkage map, perform the fine mapping the color-controlling genetic locus, and verify candidate genes using yeast two-hybrid, transcriptome analysis, and different light treatments. The results of the study indicated that the color trait of *A. cornea* is controlled by two pairs of alleles. When both pairs of loci are dominant, the fruiting body is purple, while when both pairs of loci are recessive or one pair of loci is recessive, the fruiting body is white. Based on the linkage map, the study finely mapped the color locus within Contig9_29,619bp-53,463bp in the *A. cornea* genome and successfully predicted the color-controlling locus gene A18078 (*AcveA*), which belongs to the Velvet factor family protein and has a conserved structure domain of the VeA protein. It can form a dimer with the VelB protein to inhibit pigment synthesis in filamentous fungi. Lastly, the study validated the interaction between AcVeA and VelB (AcVelB) in *A. cornea* at the gene, protein, and phenotype levels, revealing the mechanism of inhibition of pigment synthesis in *A. cornea*. Under dark conditions, dimerization occurs, allowing it to enter the nucleus and inhibit pigment synthesis, leading to a lighter fruiting body color. However, under light conditions, the dimer content is low and cannot enter the nucleus to inhibit pigment synthesis. In summary, this study clarified the mechanism of white strain formation in *A. cornea*, which could aid in improving white strains of *A. cornea* and studying the genetic basis of color in other fungi.

## 1. Introduction

The color of the fruiting bodies (FBs) of an edible fungus is determined by pigments produced during development. Color is also an essential marker for genetic research on germplasm for the breeding of varieties of edible fungi. However, relatively few studies have investigated the inheritance patterns of genes that regulate the color of the FBs of edible fungi. In addition, there is no consensus on whether the inheritance of qualitative traits, such as color, in edible fungi is controlled by a single locus or multiple loci among different varieties or even within the same variety [1,2,3,4]. For example, bulked segregant analysis and comparative transcriptome analysis based on high-quality genomes identified cytochrome P450 as a candidate gene for melanogenesis in the edible medicinal mushroom *Hypsizygus marmoreus* [5]. Moreover, bulked segregant analysis and random amplified polymorphic DNA of seven progeny groups generated from white and dark grayish brown parental strains of *Pleurotus cornucopiae* demonstrated that color was controlled by two pairs of genes [6]. However, a standardized and efficient method is needed for in-depth studies of the mechanisms that regulate pigment synthesis and the inheritance of color in edible fungi.

*Auricularia cornea* is an important edible and medicinal bipolar heterothallic fungus that is cultivated worldwide [7,8,9,10]. A white mutant of *A. cornea* that occurs by gene mutation both naturally and when cultivated is preferred by consumers. However, production of the white mutant of *A. cornea* is especially difficult and time-consuming [11]. Therefore, it is critical to elucidate the mechanisms regulating the inheritance of color to efficiently produce the white mutant strain of *A. cornea*.

A previous study [12] demonstrated that purple coloration was a dominant trait, while white coloration was a recessive trait. The authors genotyped 87 monokaryotic strains of F2 hybrids obtained by cross-hybridization, generated a genetic linkage map, and identified molecular markers based on simple sequence repeats (SSRs) and sequence-related amplified polymorphisms. The key enzyme influencing pigment synthesis was identified as glutamine-dependent amidotransferase (Gn-AT), which was also hypothesized to be the key enzyme for the synthesis of the pigment γ-glutamine-4-hydroxy-benzoate (GHB).

Although the key genes for *A. cornea* pigment synthesis have been identified, it remains unclear whether the coloration of the FBs is determined by a mutation to a single locus or multiple loci and the influence on pigment synthesis. Thus, based on previous studies of the coloration of FBs of various populations, this study mapped the loci that control color using map-based cloning. The functions of genes that control color and the pigment synthesis mechanism of *A. cornea* were verified by comparative transcriptome analysis with a yeast two-hybrid system. The aim of the present study was to improve the breeding of the white mutants of *A. cornea* and provide references for future research on color inheritance in edible fungi.

## 2. Materials and Methods

### 2.1. Test Strains

The parental strains included the white strain ACW001-33, which is recessive homozygous for color, and the purple strain ACP004-33, which is dominant homozygous for color (Figure 1). The parental monokaryotic strains ACW001-33 and ACP004-33 were collected and hybridized to obtain the hybrid strain ACW001-33ACP004-33. The mapping population consisted of 87 monokaryotic strains derived from ACW001-33 × ACP004-33.

The test-cross population consisted of 87 strains obtained by hybridization of the mapping population with ACW001-9, a monokaryotic strain of ACW001, with a different mating-type of strain ACW001-33. The mapping population was divided into two types based on the test-cross population: purple-type (P) monokaryotic strains, which produced purple FBs, and white-type (W) monokaryotic strains, which produced white FBs. 

### 2.2. Media

Potato dextrose agar medium was composed of 200 g of potato, 20 g of agar, 20 g of glucose, and 1000 mL of water.

The culture medium (300 g) consisted of 78% sawdust, 20% wheat bran, 1% gypsum, and 1% lime, which was mixed with water at a ratio of 40:60.

*Escherichia coli* was cultured with lysogeny broth (LB) medium (0.5% yeast extract, 1% tryptone, and 1% NaCl, pH 7.0), LBA medium (LB medium containing ampicillin at 100 µg/mL), and LBK medium (LB medium containing kanamycin at 50 µg/mL).

Yeast complete medium was composed of 1% yeast extract, 2% tryptone, 2% glucose, and 0.02% adenine. 

Yeast nutrient-deficient screening medium was prepared as described in the Yeast Protocols Handbook (PT3024–1) published by Clontech Laboratories, Inc. (Mountain View, CA, USA).

### 2.3. Preparation of Genetic Population

The back-cross population of dikaryon strains was generated by monohybrid crossing of the mapping population, which consisted of 87 strains, with the parental monokaryotic strains ACW001-33 and ACP004-33, respectively. Each combination was repeated three times. The back-cross population was used to determine the mating type of the mapping population. The mating types of ACW001-33 and ACP004-33 were identified as *A_w_*_1_ and *A_p_*_1_.

The selfing population of ACW001-33×ACP004-33 consisted of monokaryon strains with mating types of *A_w_*_1_ and *A_p_*_1_ that were randomly selected from the mapping population and crossed in pairs. Each combination was repeated three times. The population was used to determine the monokaryon strains with recombined loci controlling the color and the color inheritance patterns of *A. cornea*.

The recombinant inbred population consisted of dikaryon strains with recombinant color loci obtained via a fruiting test of the inbred population. Recombinant monokaryon strains with mating types of *A_w_*_1_ and *A_p_*_1_ were hybridized with monokaryon strains from the mapping population via mono–mono crossing to produce the recombinant inbred population. Each combination was repeated three times. The dikaryon strains were used to investigate the composition and location of loci controlling the color of the *A. cornea* FBs.

### 2.4. Fine Mapping of Genes That Control Color

SSRs were obtained by genome scanning using the SciRoKo 3.4 software (https://kofler.or.at/bioinformatics/SciRoKo/, accessed on 15 April 2021) with 108 pairs of primers designed and synthesized based on the flanking sequences of the SSRs (Appendix A, genome accession number PRJNA943604) [13]. Using parental ACW001-33 and ACP004-33 as templates, the specific SSR primers were selected by PCR amplification and polyacrylamide gel electrophoresis and genotyped for 87 targeted strains. The same bands as parent ACW001-33 are marked with “A”, and the same bands as parent ACP004-33 are marked with “B”. The genetic linkage of the color and molecular marker loci was mapped using the JoinMap 4.0 software (https://www.kyazma.nl/index.php/JoinMap/, accessed on 21 May 2021) with a threshold limit of detection of 3. The ACW001-33 genome contig, which contained SSR molecular markers closely related to the color locus, was identified by genetic linkage [14]. The Primer3 tool (https://primer3.org/, accessed on 27 May 2021) was used to design primers for fine mapping to differentiate the ACW001-33 and ACP004-33 genotypes [15], named TGP-X.

### 2.5. Function Prediction of Candidate Genes

The Softberry software 1.0 (http://www.softberry.com/, accessed on 13 August 2021) was used to predict the sequence of the fine-mapping region, identify candidate protein-coding genes with the *A. bisporus* genome as a reference, and then preliminarily assess the structures and functions of the candidate protein-coding genes by comparing the conserved domains with references retrieved from the National Center for Biotechnology Information database (https://www.ncbi.nlm.nih.gov/, accessed on 13 August 2021).

### 2.6. Functional Verification of Candidate Genes

Preparation of a yeast two-hybrid system for functional verification of the candidate genes: Specific primers to amplify the complementary DNA (cDNA) were designed based on the genome sequence and predicted coding sequences. The results were compared to the genome sequence to determine the full length of the candidate genes and the coding sequences. The coding sequence of each gene was determined with the yeast two-hybrid pGADT7 and pGBKT7 vectors. The yeast strain AH109 was co-transformed. Four colonies of yeast transformants were randomly chosen for self-activation. The reporter genes HIS3, ADE2, and MEL1 were detected using plate culture, and the transformants were transferred to synthetic defined (SD)/–Leu/–Trp, SD/–Ade/–His/–Leu/–Trp+X–α–Gal plates and incubated at a constant temperature of 30 °C for 4 days to promote growth. Blue coloration of the clone indicated interactions among the proteins. Otherwise, the clone was considered a false positive [16].

Transcriptome analysis: Total RNA was extracted at three different developmental stages (mycelium, primordium, and FBs) of strains ACW001 and ACP004 and sequenced. Raw reads were filtered with the fastp data pre-processing tool to obtain high-quality clean data, which were aligned to the ACW001-33 genome with the HISAT2 alignment program [17]. Based on the alignment results, transcripts were reconstructed using the StringTie algorithm and gene expression levels were calculated using the RSEM tool for the quantification of RNA-seq data [18,19]. 

Effect of light on fruiting: The hybrid strain AC31 was cultured in a fruiting chamber with varying degrees of light intensity, from full light to the complete absence of light. The experiment was repeated three times for each group and the coloration of the FBs was recorded. 

Verification by real-time quantitative polymerase chain reaction (RT-qPCR): The transcriptome data and coloration of the FBs of the AC31 strains were verified by RT-qPCR. First, total RNA was extracted using TRIzol reagent (Thermo Fisher Scientific, Waltham, MA, USA) and reverse-transcribed into cDNA using TransScript^®^ All-in-One First-Strand cDNA Synthesis SuperMix for qPCR (TransGen Biotech Co., Ltd., Beijing, China) with primers for selected differentially expressed genes (Appendix A), which were designed with the Primer3 tool (https://primer3.org/, accessed on 20 October 2021). The RT-qPCR reactions were performed with TransStart^®^ Top Green qPCR SuperMix (TransGen Biotech Co., Ltd.). Gene expression levels were calculated with the quantitative 2^–ΔΔCt^ method with *β–TUB* as an endogenous reference gene.

## 3. Results

### 3.1. Coloration of the FBs of A. cornea Is Not Controlled by a Single Locus

According to Mendel’s Laws of Inheritance, if color is determined by a single locus, the theoretical ratio of purple to white strains in the test-cross and back-cross populations should be 1:1. However, mating of 33 of the 87 back-cross populations (*A_P_*_1_) with strain ACW001-33 yielded 25 white strains and 8 purple strains (Appendix A), with a white:purple ratio of 3:1. Mating of the other 54 strains (*A_w_*_1_) with strain ACP004-33 yielded all purple strains. Thus, it was not possible to determine the color genotype of the corresponding monokaryotic strains. Additionally, the 87 strains of the test-cross population (Hybridization of 87 Mapping Populations with Test Cross Strain ACW001-9) yielded 57 white strains and 30 purple strains, with a white:purple ratio of 2:1. Neither of these two conditions conformed to the segregation ratio of 1:1, indicating that the coloration of the FBs of *A. cornea* is not controlled by a single locus.

Comparisons of the FB color of the back-cross and test-cross populations found that the FBs of monospores 2, 4, and 23 produced by test-crossing of the mapping population with strain ACW001-9 were purple, while those back-crossed with strain ACW001-33 were white. This phenomenon indicates that ACW001 is homozygous for the white trait, but not for the color locus composition. Therefore, further investigations were required.

### 3.2. Acquisition of Recombinant Monokaryotic Strains

In the mapping population, inbreeding of 29 monokaryon strains with known mating and color types yielded 141 (67.1%) of 210 inbred strains that produced FBs (Appendix A). With the exception of the inbred monokaryotic strain nos. 7 (*A_w_*_1_, P), 23 (*A_p_*_1_, P), and 51 (*A_w_*_1_, P), the other inbred strains adhered to the following rules: P and P generate purple FBs; P and W generate purple FBs; and W and W generate white FBs. However, when mated with W-type or P-type monokaryotic strains, nos. 7, 23, and 51 may produce purple or white FBs because coloration is controlled by multiple pairs of recombined loci. Therefore, it was determined that strain nos. 7, 23, and 51 were monokaryotic strains with recombinant color loci.

### 3.3. Fine Color Typing of the Mapping Population

To elucidate the mechanism underlying the above phenomenon, monokaryotic strain nos. 23 and 51 were hybridized with the mapping population to construct a recombinant inbred population. The two strains had different mating types and could successfully mate with all of the monokaryotic strains of the mapping population. In total, 61 strains were obtained from the recombinant inbred population, which included 30 strains that produced white FBs and 31 that produced purple FBs. Strain nos. 23 and 51 were monokaryotic strains with the recombinant purple type and assumed as the P_r_ type. Based on the color of FBs after hybridization with strain nos. 23 and 51, the monokaryotic strains in the mapping population can be further divided into the following four types: (1) non-recombinant purple (P_d_) type, where the P type produced purple FBs after mating with no. 23 or 51 (14 strains in total); (2) recombinant purple (P_r_) type, where the P type produced white FBs after mating with no. 23 or 51 (7 strains in total); (3) non-recombinant white (W_d_) type, where the W type produced white FBs after mating with no. 23 or 51 (24 strains in total); and (4) recombinant white (W_r_) type, where the W type produced purple FBs after mating with no. 23 or 51 (16 strains in total) (Appendix A).

### 3.4. Coloration of the FBs of A. cornea Is Controlled by Two Loci

The coloration of the FBs obtained by mating among the four types of monokaryotic strains conformed to the following rules: the hybrid FBs of P_d_ × P_d_ were purple; P_d_ × P_r_ were purple; P_d_ × W_d_ were purple; P_d_ × W_r_ were purple; P_r_ × W_r_ were purple; W_d_ × W_d_ were white; W_d_ × W_r_ were white; W_d_ × P_r_ were white; W_r_ × W_r_ were white; and P_r_ × P_r_ were white (Table 1). The ratio of purple to white FBs was 9:7, consistent with the segregation ratio of the same trait controlled by two pairs of alleles. Therefore, it can be inferred that the pigment synthesis of *A. cornea* is controlled by two pairs of alleles, named loci A and B.

### 3.5. Analysis of Color Locus Composition

The pigment synthesis of *A. cornea* requires the participation of loci A and B. Only when the two alleles of inbred strains are dominant (i.e., A_B_), the FBs are purple, and when both alleles are recessive (i.e., aabb) or only one allele is dominant (i.e., A_bb or aaB_), the FBs are white.

The inbred FBs of P_d_ and all other types of strains were purple; thus, the locus type of P_d_ was the same as that of the purple monokaryotic parent ACP004-33, which was the AB type. The FBs of W_d_ and P_d_ were purple, while those of the other three types of strains were white. Therefore, the locus type of W_d_ was the same as that of white monokaryotic parent ACW001-33, which was the ab type. The inbred FBs of P_r_ × P_d_ and W_r_ × P_d_ were purple, while those of P_r_ × W_d_, W_r_ × W_d_, W_r_ × W_r_, and P_r_ × P_r_ were white. Moreover, mating of P_r_ and W_r_ produced purple FBs. These findings demonstrate that the color loci were recombined (aB and Ab types, respectively).

The FBs produced by the crossing of P_r_ monokaryotic strain nos. 2, 4, and 23 with ACW001-9 were purple, while those produced by back-crossing with ACW001-33 were white, indicating that although the monokaryotic strains were obtained from the same white parent, the color loci differed between strains ACW001-9 and ACW001-33. Although ACW001 is homozygous for the white trait, the composition of the color loci is not homozygous. The color loci of the monokaryotic strain nos. 2, 4, and 23 were the aB type, and those of strain ACW001-33 were the ab type. Therefore, the color locus of strain ACW001-9 should be the Ab type. These findings confirmed the color loci of the strains used in this study (Figure 2).

### 3.6. Fine Mapping of Candidate Genes for Color Control at Locus A

The color loci of strain nos. 23 and 51 in the mapping population were the Pr type (aB) and locus B of all 61 recombinant inbred strains was dominant. Therefore, locus A determined the color of the FBs of the recombinant inbred strains. In the mapping population, the color locus of the monokaryotic strains that produced purple FBs was the A_ type (31 strains in total) and that of the monokaryotic strains that produced white FBs was the a_ type (30 strains in total). By associating locus A with phenotypic traits, 54 SSR molecular markers were screened to locate locus A and the JoinMap 4.0 software was used to perform linkage analysis and generate a genetic linkage map of locus A. The sequences of the SSR primers are shown in Appendix A. The findings revealed that locus A was linked to the markers SSR2507 and SSR2514 (Figure 3A) and was located between 0 and 197,550 bp of genome Contig9 (Figure 3B). Based on this sequence, seven pairs of PTG-specific primers were generated for fine mapping (Appendix A), which revealed that the color control locus was located between the PTG-20 and PTG-21 loci (Figure 3C).

### 3.7. Prediction of Candidate Functional Genes for Color Control

The protein-encoding gene A18078 was mapped by gene prediction using the SoftBerry software (Figure 3D). Further analysis revealed that the candidate protein-coding gene included a Velvet conserved domain associated with the VeA protein in the Velvet factor family (pfam11754), named AcVeA. Velvet factor family proteins are commonly found in filamentous fungi [20]. The N-terminus of the VeA protein contained a Velvet domain and the C-terminus contained an unstable PEST sequence, which is a region rich in proline (P), glutamic acid (E), serine (S), and threonine (T) [21,22]. Sequence analysis demonstrated that the AcVeA protein was composed of 554 amino acids. The epestfind algorithm (https://emboss.bioinformatics.nl/cgi–bin/emboss/epestfind, accessed on 15 August 2021) was used for regional prediction. The epestfind scores range from –50 to +50, where a score >0 denotes a possible PEST region and >5 sparks real interest [23]. The results illustrated that the C-terminus of the AcVeA protein contained 12 amino acid regions with a score of +18.83, indicating the function of AcVeA (Figure 4A). The VeA and VelB proteins can form a Velvet dimer in fungi to inhibit pigment synthesis [24]. Using the SWISS–PROt database (https://www.ebi.ac.uk/uniprot/, accessed on 17 August 2021), the genome of *A*. *cornea* was predicted to include the VelB-like protein AcVelB (A13646, Contig19). The Velvet domain was confirmed by comparisons with conserved domains retrieved from the National Center for Biotechnology Information database and the sequence analysis of VelB proteins of other species with the same characteristic of the Velvet domain of being divided into two segments. The sequence analysis results showed that AcVelB is composed of 366 amino acids, with the Velvet domain separated by 123 amino acids.

### 3.8. Verification of the Functions of Candidate Genes for Color Control

Since the regulation of pigment synthesis is based on the formation of dimers by predicted candidate genes, the interaction between the two genes can be verified with the yeast two-hybrid system. The corresponding results of the *AcveA* and *AcvelB* genes showed that all AH109 strains co-transformed with vectors encoding the DNA-binding domain and transcription activation domain had produced clones on the synthetic dropout medium, thereby verifying successful co-transformation (Figure 4B). Clones on SD/–Leu/–Trp culture medium were further cultured on SD/–Ade/–His/–Leu/–Trp medium. However, clones of only strain AH109 containing the target gene plasmid were produced (Figure 4C), indicating interactions between the AcveA and AcvelB proteins.

The results of transcriptome and RT-qPCR analyses showed that the purple strain did not express *AcveA* (Figure 4D), possibly due to a sequence mutation. Comparisons of strains ACW001 and ACP004 found that *AcveA* formed the termination codon with a single base mutation that occurred at 100 bp away from the initiation codon of strain ACP004, which prematurely terminated transcription (Figure 4E). Although significantly increased in the white strain, the expression of *AcveA* was limited when exposed to light, suggesting that the expression of *AcveA* is suppressed by light, even though the FBs continue to develop, resulting in an average level of expression. The expression profile of *AcvelB* was similar to that of *AcveA* (Figure 4F). Unlike VelB, which lacks nuclear localization signals, VeA is generally involved in transport processes [24]. VeA and VelB often form a dimer in the cytoplasm, which enters the nucleus. The results of transcriptome and RT-qPCR analyses further confirmed these associations. 

Regulation of pigment synthesis by the Velvet dimer is closely related to light intensity. Strain AC31, which carries no mutation at loci A and B, was selected to test the effects of different light intensities on the coloration of the FBs. The results demonstrated that the FBs of strain AC31 were dark purple under light conditions and light pink under dark conditions (Figure 4G). The RT-qPCR results showed that the relative expression of *AcveA* was approximately four-fold greater in the dark than the light (Figure 4H). These results indicate that the AcVeA–AcVelB dimer regulates pigment synthesis in the FBs of *A*. *cornea*. 

## 4. Discussion

### 4.1. Roles of Mating Types in Genetic Populations

Similar to *Auricularia heimuer*, it remains unclear whether the mating type of *A. cornea* is bipolar or tetrapolar [25,26]. However, a previous study reported that *A. heimuer* is a bipolar heterothallic fungus via analysis of the mating-type loci using a full-round mating method in conjunction with whole-genome sequencing and genetic linkage mapping [27]. In addition, various mating experiments verified that, similar to *A*. *heimuer*, *A. cornea* employs a bipolar heterothallic mating system.

The monospore of the same earpiece of *A. cornea* has two mating types and the same mating type cannot produce fruit; thus, the success rate of inbreeding is only 50%. The mating of monokaryotic strains must be regulated to generate a large number of genetic populations for experimentation. Approximately half of the cross combinations will be unable to mate when fully adopting the random combination method in the inbred and recombinant inbred populations. Each cross combination would need a microscopic examination, which increases the workload and the chance of error. Therefore, the mating type of the collected monokaryotic strains was first identified, which successfully facilitated the predictability of the mating results and increased the efficiency and precision of this experiment.

### 4.2. Color Segregation Ratio of FBs among Different Genetic Populations and Genetic Relationship between Two Color Control Loci 

Strain ACW001 is not recessive homozygous at the color control locus A and lacks a dominant gene at the color control locus B and, thus, is not able to synthesize the key enzyme B, which results in the generation of white inbred progenies. In contrast, strain ACP004 is dominant homozygous at the color control loci A and B and produces purple inbred and hybrid progenies with strain ACW001 (Figure 5A). Alignment with the BioEdit tool (https://bioedit.software.informer.com/, accessed on 15 January 2022) revealed that the color control locus B is located on the genome Contig19, while the *AcveA* locus is located on the genome Contig9. Since the two genes are not on the same contig and there is no genome at the chromosome level, it remains unclear whether the two loci are on the same chromosome and, therefore, it is impossible to confirm the possible linkage between the loci. According to the Law of Independent Assortment, if there is no linkage between locus A and locus B, the ratio of gametes produced by the four combinations of ACW001-33×ACP004-33 should be 1:1:1:1 and the theoretical ratio of white to purple FBs should be 1:1 (Figure 5B). After back-crossing with recessive monokaryotic parental strains, the theoretical ratio of white to purple FBs was 3:1 (Figure 5C). The inbred progenies of the hybrid ACW001-33×ACP004-33 were not completely produced by random pairing due to restrictions of the mating-type genes; thus, the color segregation ratio was not considered in this study.

The results showed that 25 and 8 strains produced white and purple FBs in the back-cross population of recessive monokaryotic parents, respectively, with a theoretical segregation ratio close to the non-linkage loci of 3:1. However, 57 and 30 strains produced white and purple FBs in the test-cross population, respectively, at a ratio of around 2:1, which is far from the theoretical ratio of 1:1. In addition, the proportion of recombinant gametes of color control loci A and B during the meiosis of ACW001-33×ACP004-33 was 37.7%, which is close to 50%, rendering it difficult to determine whether loci A and B are on the same chromosome. Therefore, when plotting the color control loci A and B of different strains or genetic populations, it is assumed that loci A and B are on the same chromosome with a linkage relationship.

### 4.3. Distorted Segregation of Mating-Type Loci and Color Loci

Distorted segregation refers to a deviation from the expected Mendelian ratio in the observed phenotypic segregation or genotype ratio. Analysis based on traditional genetic concepts and techniques will lead to variation or even inaccuracy [28]. Many segregation loci have been discovered in different food and vegetable crops since Mangelsdorf et al. first reported distorted segregation in maize in 1926 [29]. Use of the chi-square test determined that the monokaryotic mating-type loci of the hybrid ACW001-33×ACP004-33 (*A_w_*_1_:*A_p_*_1_ = 54:33) and color loci of the test-cross population (_B:b = 30:57) exhibited distorted segregation, which did not follow the Mendelian ratio. As a feasible explanation, the segregation of locus B led to the difference in the color segregation ratio of the FBs in the test-cross population and the theoretical value of 1:1. The primary causes of distorted segregation reportedly include (1) a recessive lethal mutation carried by an allele [30]; (2) genetic relationships, as the farther the genetic relationship, the greater the proportion of distorted segregation [31]; (3) subtle influences of the parental cytoplasmic genome [32]; and (4) structural rearrangements, deletions, insertions, and mutations of chromosomes during hybridization [33]. However, the reasons for the distorted segregation of the test-cross population in the present study are unclear, thus warranting further investigations. 

### 4.4. Regulatory Mechanism of Velvet Family in the Pigment Synthesis Pathway

The Velvet protein family includes four members: VeA, VelB, VosA, and VelC [34]. The formation of heterodimers and homodimers, as well as the subsequent formation of trimers with LaeA proteins, are the primary mechanisms used by Velvet family proteins, although quite complex, to regulate the secondary metabolism and development of fungi. A previous investigation of the *A*. *flavus* genome discovered that numerous secondary metabolites of polyketides, as the source of fungal pigments, are influenced by VeA [35,36], indicating the role of VeA in pigment regulation. Meanwhile, *velB* is reported to prevent the formation of pigments [37,38]. The production of melanin is dramatically increased in *Botrytis cinerea* following the deletion of *velB* and the expression levels of gene clusters involved in pigment synthesis are significantly increased [39]. Deletion of *velB* in *Fusarium graminearum* is reported to influence the ability to synthesize pigments due to the subsequently decreased expression of *PKS12*, *Gip1*, and *Gip2*, which are involved in pigment production, and the increased expression of *AurJ*, *AurF*, *AurO*, and *AurR* [40]. Similarly, deletion of *velB* in *A. nidulans* promotes the synthesis of a brown pigment [41]. VeA and VelB form dimers in the cytoplasm, which enter the nucleus and combine with the protein LaeA to form a Velvet trimer complex through the S-adenosylmethionine locus, thus participating in the regulation of secondary metabolism (Figure 6). However, LaeA was not a functional gene predicted by the *A. cornea* genome and transcriptome, suggesting that the Velvet dimer can inhibit pigment production without the participation of LaeA. The Velvet family proteins also control various secondary metabolic pathways in addition to growth and development. Hence, further studies are needed to determine the precise roles of Velvet family proteins, as well as the influence of LaeA deletion. 

### 4.5. Mechanistic Differences in Pigment Synthesis Regulated by the A and B Loci

Genotyping and gene mapping based on the test-cross between ACW001-9 and the mapping population found that the FB color was regulated by Gn-AT [12]. The color type of ACW001-9 was Ab, which leads to the dominance of locus A in the test-cross population, and the color of the test-cross population is controlled by locus B. Therefore, the regulatory gene at locus B is Gn-AT, which plays a role in catalyzing the formation of para–aminobenzoic acid from branching acids in the GHB pigment synthesis pathway. Additionally, the regulatory gene at locus B was amplified in all purple, but not white, dikaryon strains, indicating no regulatory gene at locus B in white strains. Therefore, pigment synthesis and the metabolic pathway were disrupted due to the deletion of a critical enzyme gene at locus B, resulting in white FBs. Since locus B can only change the FB color between white and purple, it should be considered more of a qualitative trait. 

Through a change in the expression level, *AcveA* at locus A can alter the color of FBs. VeA is reported to decrease the expression of the pigment gene by 98% in *F. graminearum*, indicating that VeA could almost entirely inhibit pigment synthesis [40], suggesting that locus A can alter the color of FBs from purple to white, as well as the degree of purple depending on the expression level of AcVeA–AcVelB. Quantitative traits are characterized by such constant changes. However, the expression profiles of many important regulatory genes in the synthetic pathway of GHB pigment or the regulatory pathway of the Velvet dimer remain unknown. Therefore, further research is warranted to elucidate the mechanism underlying the color inheritance of the FBs of *A. cornea*. 

### 4.6. Positional Cloning Technology Based on Genetic Linkage Maps for Mapping of Functional Genes

Since the invention of positional cloning technology based on genetic linkage maps [42], it has played a vital role in the discovery of functional genes and mutation sites in plants [43]. Using this method, both Wang [12] and this study successfully mapped the color-controlling locus gene in *A. cornea*. Compared with plants, fungal gene mapping has more advantages as fungal genomes are small, with fewer redundant sequences, making the mapping process more efficient. Additionally, haploid progeny produced by fungi during meiosis can be directly cultured, allowing for the rapid acquisition of localized populations that are similar to haploid-doubling populations with the highest mapping accuracy in plants. Furthermore, fungi have short culture periods, which can shorten the experiment cycle. However, the pathway of pigment synthesis is very complex and controlled by multiple genes. Classic genetics combined with molecular marker methods cannot be used to identify the complete pathway or the expression of the main-effect genes. Therefore, it is necessary to use multi-omics association analysis to decipher the complete mechanism of pigment synthesis and regulation in *A. cornea*.

## Figures and Tables

**Figure 1 jof-09-00412-f001:**
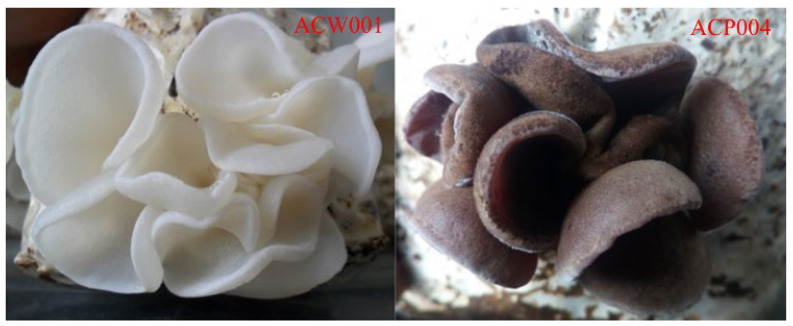
FBs of test strains.

**Figure 2 jof-09-00412-f002:**
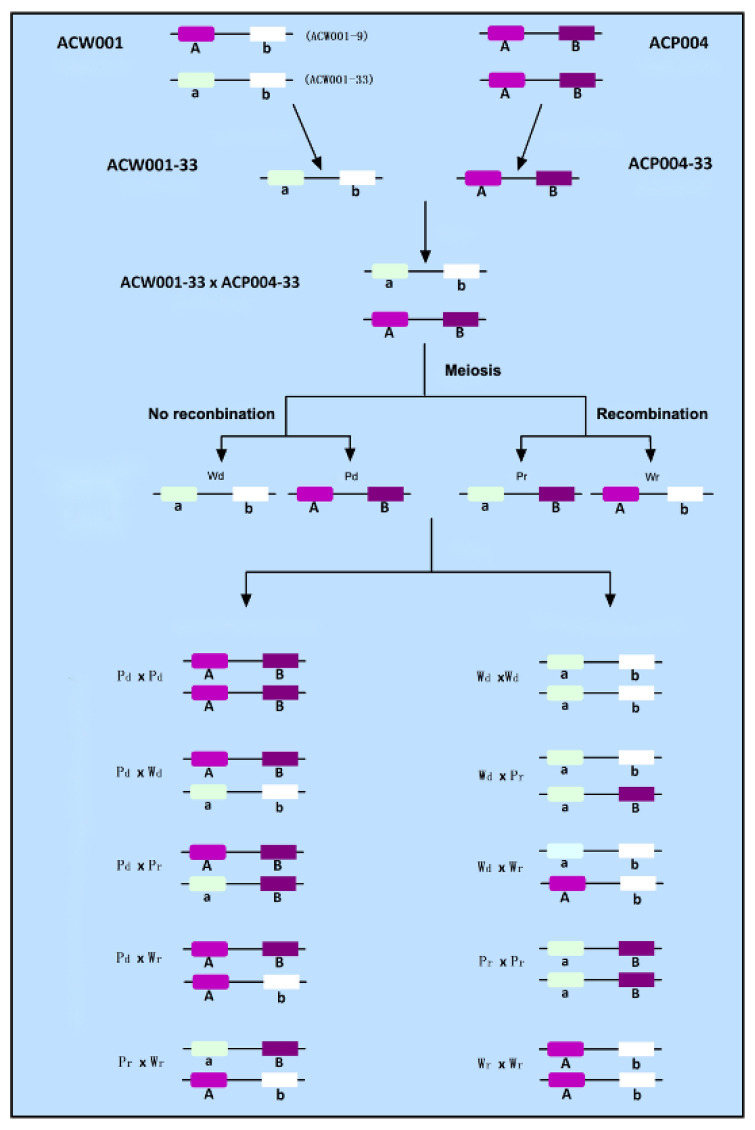
Composition of the color loci of *A. cornea* strains. Use A(a) and B(b) to represent two color loci. This map is only suitable for two color loci located on the same chromosome with linkage. If the two color loci are located on different chromosomes, the A and B loci in the figure would be disconnected.

**Figure 3 jof-09-00412-f003:**
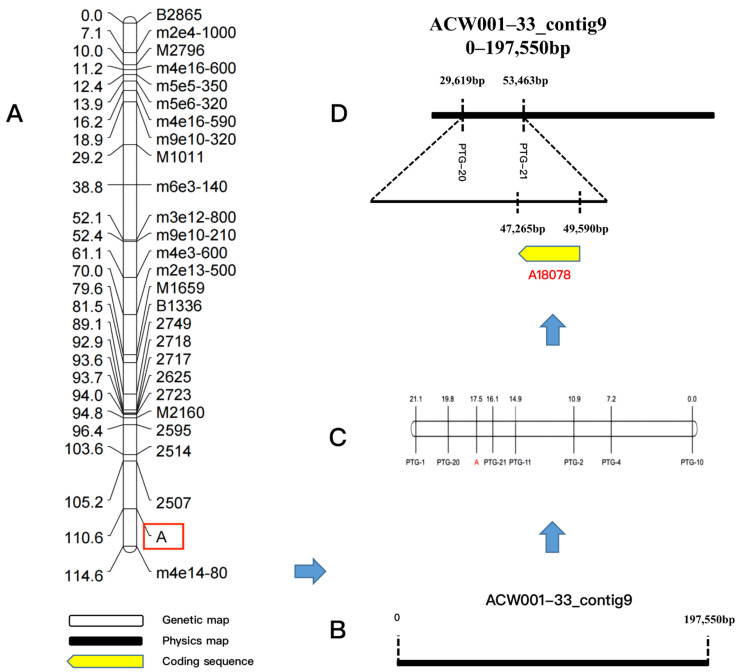
Fine mapping of the color control locus and candidate genes. (**A**): Preliminary location linkage map of the color control locus, in the figure, A is the color control loci; (**B**): Physical map of ACW001–33_contig9; (**C**): Fine mapping linkage map of the color control locus by PTG-specific primers; (**D**): Location of the color control locus in the ACW001–33 genome and prediction of functional genes in this region.

**Figure 4 jof-09-00412-f004:**
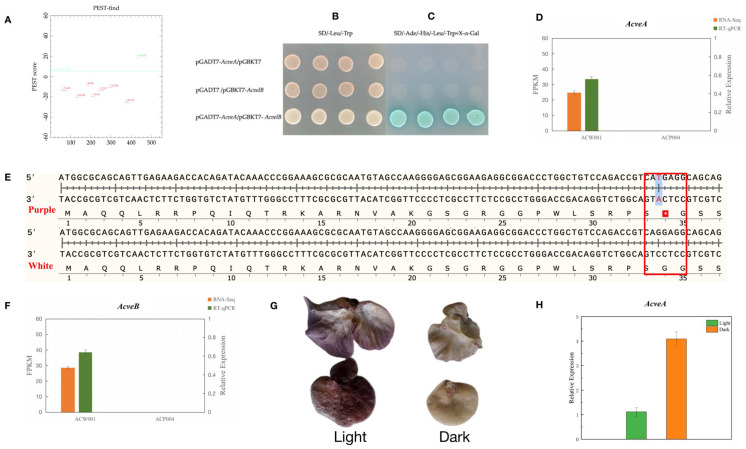
Prediction of VeA function. (**A**): The epestfind prediction of VeA; (**B**,**C**): Validation of the interactions between AcveA and AcvelB with a yeast two−hybrid system. (**D**): RT−qPCR results of AcveA of the two parental strains, The red box represents the SNP mutation region. (**E**): Sequence comparison of *AcveA* gene between purple and white strains−the mutation of guanine into thymine causes early termination of transcription. (**F**): RT−qPCR results of AcvelB of the two parental strains. (**G**): Color of the FBs of strain AC31 under light and dark conditions. (**H**): Histogram of the relative expression of AcveA in AC31 strains with different colors.

**Figure 5 jof-09-00412-f005:**
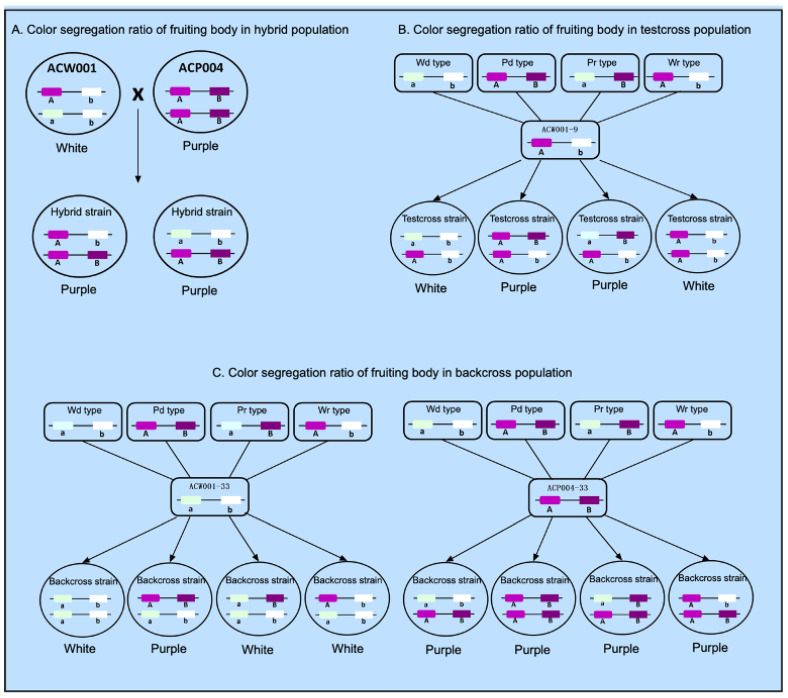
Color segregation ratio of FBs of different genetic populations of *A. cornea*. Use A(a) and B(b) to represent two color loci. This map is only suitable for two color loci located on the same chromosome with linkage. If the two color loci are located on different chromosomes, the A and B loci in the figure are disconnected.

**Figure 6 jof-09-00412-f006:**
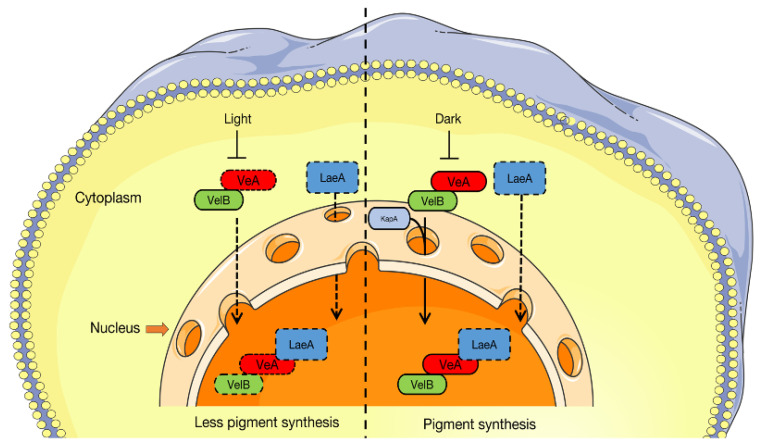
A model of VeA supporting nuclear localization of VelB and formation of the Velvet complex. The left side shows the damage of the Velvet family transport under light conditions; the dotted lines indicate the decreased amount of VeA in the cell and the impairment of VeA–VelB nuclear transport in the light. On the right is the process of the Velvet complex entering the nucleus under dark conditions. LaeA is not found in the *A. cornea* genome and, thus, is represented by dotted lines.

**Table 1 jof-09-00412-t001:** FB color of hybridized monokaryotic strains.

Type of Monokaryon	P_d_	P_r_	W_d_	W_r_
P_d_	P	P	P	P
P_r_	P	W	W	P
W_d_	P	W	W	W
W_r_	P	P	W	W

## Data Availability

Not applicable.

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
