# Peer review of "Velvet Family Members Regulate Pigment Synthesis of the Fruiting Bodies of Auricularia cornea"

_jof, 2023, doi:10.3390/jof9040412_

Round 1

Reviewer 1 Report

In this study, there are two strains of A, cornea, white mutant and purple wild type which were crossed.  These backcrosses were mated with the white strain and the daughters were evaluated in terms of physical genetic map by random amplification, which identified whether the section originated form the mutant or wild type.  Based on the data from the different strains it was possible to identify the appropriate fragments containing the desired genes. An RNA profile was created and then PCR primers were designed from those that were expressed.  Several genes were identified that were involved colour formation that worked indendently, with one suppressing colour formation. 

The abstract contains some background information rather than results

Line 51 this must be the wrong citation

Line 99 why the need to repeat three times?

line 104 was

Line 115 is there a genome  sequence of A. cornea?
Does the amplification of the parental strains provide a unique pattern - this is  not clear

Line 126 what is PTG? it is not clear how these primers were designed

line 164  what is being amplified?

line 177 confused about this mating

line 243 italics

Not sure whether the authors would agree to a conclusions section

Author Response

English language editing has been refined in accordance with MDPI recommendations.

Line 1 The overall abstract has been modified

Line 58 The incorrect citation was changed。

Line 106 The purpose of three repetitions is to ensure the accuracy of the data.

Line 124 The SSR primer sequence was added to Table S1. The genome sequence has been uploaded to NCBI , genome accession numbers PRJNA943604。

Line 125 Modify to “Using parental ACW001-33 and ACP004-33 as templates, the specific SSR primers were selected by PCR amplification and polyacrylamide gel electrophoresis and genotyped for 87 targeted strains. The same bands as parent ACW001-33 are marked with "A", and the same bands as parent ACP004-33 are marked with "B".”

Line 132 Modify to “The Primer3 tool (https://primer3.org/) was used to design primers for fine mapping to differentiate the ACW001-33 and ACP004-33 genotypes, named TGP-X.”

Line 184 The test-cross population consisted of 87 strains obtained by hybridization of the mapping population with ACW001-9, The information of ACW001-9 is described in 2.1.

Line 252 “A.cornea” has been modified to italic

Please see the attachmentEnglish Editing Certificate

Reviewer 2 Report

The authors proved that the coloration of the fruiting bodies (FBs) of A. cornea was jointly controlled by two alleles. VeA-VelB dimer was proven to be involved in the pigment synthesis of the A. cornea FBs by a yeast two-hybrid assays, transcriptome analysis, and different light treatments. The experiments were conducted in a logic and proper way and lead to the planned results. However, there are some issues that should be addressed.

1. Line 300, comparisons of strains ACW001 and ACP004 found that AcveA formed the termination codon with a single base mutation that occurred at 100 bp away from the initiation codon of strain ACP004. It is recommended to present the results of the single base mutation in the form of sequence alignment in the manuscript.

2. The pigment of A. cornea has been identified according to previous literature, so it is suggested to supplement the relevant research progress in the Introduction. In addition, the relevant discussion based on the results of this manuscript should be added based.

3. The caption of Figure 6 should be enriched to provide more information.

4. Have the authors clearly emphasized the strengths of their study/theory/methods/argument?

5. Have the authors clearly stated the limitations of their study/theory/methods/argument?

6. Some degree symbol is not proper (symbol option can be used from microsoft word).

7. References: I suggest replacing old references with the latest ones.

Author Response

English Editing CertificateEnglish language editing has been refined in accordance with MDPI recommendations.

Line 310 The sequence comparison results of the AcveA gene between the white strain and the purple strain have been added in Figure 4-E.

Line 58 Research Progress on the Pigment of Auricularia cornea. The relevant discussion based on this result is on line 428

Line 423 Enriches the caption content of Figure 6

Line 449 Advantages and limitations of research methods were added to the discussion

Corrected the use of symbols
